# DLSMR: Deep Learning-Based Secure Multicast Routing Protocol against Wormhole Attack in Flying Ad Hoc Networks with Cell-Free Massive Multiple-Input Multiple-Output [note 1]

**DOI:** 10.3390/s23187960

**Published:** 2023-09-18

**Authors:** Yushintia Pramitarini, Ridho Hendra Yoga Perdana, Kyusung Shim, Beongku An

**Affiliations:** 1Departement of Software and Communications Engineering in Graduate School, Hongik University, Sejong City 30016, Republic of Korea; yushintia@mail.hongik.ac.kr (Y.P.); hendra@mail.hongik.ac.kr (R.H.Y.P.); 2School of Computer Engineering & Applied Mathematics, Hankyong National University, Anseong City 17579, Republic of Korea; kyusung.shim@hknu.ac.kr; 3Departement of Software and Communications Engineering, Hongik University, Sejong City 30016, Republic of Korea

**Keywords:** CF-mMIMO, clustering, deep learning, flying ad hoc networks, secure multicast routing, security, wormhole attack

## Abstract

The network area is extended from ground to air. In order to efficiently manage various kinds of nodes, new network paradigms are needed such as cell-free massive multiple-input multiple-output (CF-mMIMO). Additionally, security is also considered as one of the important quality-of-services (QoS) parameters in future networks. Thus, in this paper, we propose a novel deep learning-based secure multicast routing protocol (DLSMR) in flying ad hoc networks (FANETs) with cell-free massive MIMO (CF-mMIMO). We consider the problem of wormhole attacks in the multicast routing process. To tackle this problem, we propose the DLSMR protocol, which utilizes a deep learning (DL) approach to predict the secure and unsecured route based on node ID, distance, destination sequence, hop count, and energy to avoid wormhole attacks. This work also addresses key concerns in FANETs such as security, scalability, and stability. The main contributions of this paper are as follows: (1) We propose a deep learning-based secure multicast routing protocol (DLSMR) to establish a high-stability multicast tree and improve security performance against wormhole attacks. In more detail, the DLSMR protocol predicts whether the route is secure based on network information such as node ID, distance, destination sequence, hop count, and remaining energy or not. (2) To improve the node connectivity and manage multicast members, we propose a top-down particle swarm optimization-based clustering (TD-PSO) protocol to maximize the cost function considering node degree, cosine similarity, cosine distance, and cluster head energy to guarantee convergence to the global optima. Thus, the TD-PSO protocol provides more strong connectivity. (3) Performance evaluations verify the proposed routing protocol establishes a secure route by avoiding wormhole attacks as well as by providing strong connectivity. The TD-PSO clustering supports connectivity to enhance network performance. In addition, we exploit the impact of the mobility model on the network metrics such as packet delivery ratio, routing delay, control overhead, packet loss ratio, and number of packet losses.

## 1. Introduction

Unmanned aerial vehicle (UAV) communications play an important role in modern network infrastructure, particularly in flying ad hoc networks (FANETs) integrated with cell-free massive multiple-input multiple-output (CF-mMIMO) [1,2]. These modern communications setups rely on effective routing methods which have a direct impact on performance and efficiency. Recently, security has emerged as a significant concern, warranting special attention to prevent potential breaches [3,4,5,6,7,8].

Given its airborne position, the UAV can cover large areas of the ground, providing superior network coverage over traditional ground-mounted access points (APs) [9,10]. Therefore, clustering or member management is very important, ensuring optimal utilization of the UAV’s wide range of capabilities and maintaining network efficiency [11,12]. One of the major security threats to these networks is wormhole attacks which are characterized by capturing enemy nodes and tunnelling packets to other locations on the network [13]. This misleading tactic breaks regular data transmission pathways and modifies network topology causing substantial network performance and reliability disruptions. Consequently, understanding these attacks and developing robust defenses against them is essential to ensure secure and reliable UAV communications.

A multicast routing protocol is a key technique in ensuring reliable data communication to multiple destinations [14]. The multicast routing protocol can select the best next node to establish the optimal route from the source to the multiple destinations [15]. Particularly in the FANET environment, UAVs move dynamically so that there can be a risk of broken connectivity and attack from eavesdroppers [16]. The authors of [16] proposed a method to detect a black hole in a short time. The authors of [17] studied a hybrid authentication scheme with a digital signature to improve the security performance in the UAV and ground node against wormhole attacks. However, the proposed method has a high processing time to authenticate the secure node.

In this context, deep learning models present an intriguing opportunity [18,19]. When the deep learning models are applied to FANETs, these models can offer sophisticated route verification mechanisms. These mechanisms predict the next secure node in the route, ensuring that the chosen path is free from wormholes before initiating data transmission. The use of deep learning to improve security in UAV communications forms the basis of this paper.

### 1.1. Related Works

Some works have been studied to improve network stability in FANETs. The authors of [20] proposed bio-inspired clustering for the FANET environment. They develop a bio-inspired clustering protocol to improve energy efficiency (EE) and manage UAV mobility. However, it does not consider safety considerations to reduce collisions and enhance stability. In [21], the authors proposed an intelligent cluster routing scheme (CRSF) to address UAV communication issues. Additionally, the CRSF can improve the stability performance in UAV communication. Nevertheless, the CRSF has a high control overhead during the clustering process. The authors of [22] used clustering as a solution to improve network stability. In addition, they also studied EE localization and clustering for UAV wildfire monitoring. The authors of [23] proposed a down–up particle swarm optimization to improve the scalability of the FANET system. However, this work only considered the optimization approach to improve stability performance.

Regarding secure multicast routing protocols, improved security performance in the FANET system was studied in [24]. Scalable and predictive routing (SP-GMRF) was proposed to predict the next node. Based on node position, the SP-GMRF provided the shortest distance to each destination node. However, SP-GMRF does not consider the clustering protocol, so it has a high control overhead. The authors of [25] proposed a distributed tree-based routing (DPTR) for FANETs, forming a network that avoids fragmentation. In this approach, a tree-like structure connects all the UAVs in the networks. Nevertheless, the routing delay is high since every node calculates the entire routing tree. In [26], the authors studied swarm EE multicast routing (SEMRP) for UAV swarms for UAVs in group formation. The SEMRP can reduce packet loss and packet re-transmission, and delay performance. However, the SEMRP has a high control overhead during the routing process. The authors of [27] proposed a credible neighbor discovery (CRFNE) algorithm to shield messages against wormhole attacks in wireless sensor networks (WSNs). The CRFNE can detect wormhole attacks during route discovery. However, the CRFNE has a high processing time due to the greater number of steps to calculate the wormhole threshold to detect the wormhole node. Based on the AODV routing protocol, the authors of [28] designed a wormhole-immune routing protocol. However, the DAWA protocol has a high control overhead because the DAWA protocol broadcasts the packet discovery to all nodes in the network. Meanwhile, the authors of [29] proposed a hybrid price auction (HPA)-based routing protocol to avoid sinkhole attacks. In addition to that, the HPA protocol can improve the security, routing delay, and scalability performance. Nevertheless, they only consider sinkhole attacks in the network. On the other hand, the DL is applied to improve end-to-end (E2E) delay in aeronautical ad hoc networks (AANETs) [30]. The authors of [30] explored a DL-based multicast routing protocol for mapping the local geographic information observed by the forwarding node into the information required for determining the optimal next hop. However, they have not considered wormhole attacks in the networks. Additionally, the authors of [31] studied reinforcement learning-assisted secure routing to minimize delays and map geographic information using Q learning. Thus, the proposed protocol can select the secure route efficiently. However, they did not mention which kind of attack was considered.

The above-mentioned works partially addressed the raised network issues such as secure multicast routing, network scalability, and deep learning approaches. In more detail, some studies have examined the effect of DL on secure routing [30,31]; others have studied the design of secure multicast routing against eavesdropped attack [24,30]; very few works have explored the effect of DL techniques in the secure multicast routing protocol specifically against wormhole attacks on security, scalability, and stability performance, which is a critical performance metric in 5G wireless networks.

### 1.2. Motivation and Contributions

Based on the abovementioned, secure and multicast routing in FANETs have been studied considerably. However, there is still a gap in the literature regarding secure multicast routing against wormhole attacks in FANETs with CF mMIMO networks. To fill this gap, we proposed a DL-based secure multicast routing (DLSMR) protocol against wormhole attacks that allows the protocol to predict the next secure node within a short time. Furthermore, the combination with TD-PSO-based clustering allows the protocol to find the optimal cluster header. The goal is to improve the security, scalability and stability performance, and achieve an optimal secure route that meets the requirements of 5G and beyond 5G wireless networks. The main contributions of this article can be summarized as follows:We propose a deep learning-based secure multicast routing (DLSMR) protocol to establish a high-stability multicast tree to avoid wormhole attacks in FANETs with CF-mMIMO. Specifically, we utilize a deep learning model to predict whether the next node is a wormhole or not. Additionally, we use various network parameters to establish routes that support more strong connectivity.We design a novel top-down particle swarm optimization (TD-PSO)-based clustering protocol in FANETs to reduce control overhead and improve route connectivity. The proposed TD-PSO, considering the node position, velocity, direction, and remaining energy, forms a cluster to optimize the cost function by combining the remaining energy weight, cosine similarity, cosine distance, and node degree. This strategic approach leads to electing cluster heads. Furthermore, to ensure communication continuity between cluster heads when the subsequent ones fall outside of the transmission range, our TD-PSO protocol also designates gateway nodes.The performance evaluations show that the proposed DLSMR with TD-PSO protocol can establish a more robust route against wormhole attacks than the benchmark protocol. Additionally, the proposed TD-PSO clustering supports stronger connectivity as clustering changes the network topology hierarchically. In addition to that, we also compare the proposed protocol under two different mobility models (reference point group mobility and random waypoint) to show the effectiveness of the proposed protocol.

The remainder of the paper is organized as follows. Section 2 introduces the overview of wormhole attack. Section 3 introduces the particle swarm optimization theory. Section 4 introduces the proposed routing protocol that consists of the basic concept of the proposed routing protocol, the proposed TD-PSO-based clustering protocol, the proposed DLSMR, and proposed deep-learning design. Section 5 presents the performance evaluation that consists of simulation environments and parameters, performance metrics, and numerical results. Section 6 concludes the paper. For ease of presentation, Abbreviations summarizes the main abbreviations used in this paper.

## 2. Overview of Wormhole Attack

A wormhole attack is a type of network security threat that affects wireless networks, including flying ad hoc networks (FANETs). In particular, FANETs consist of a group of unmanned aerial vehicles (UAVs) that communicate with each other to form a self-organizing and self-healing network. Due to their dynamic topology, limited resources, and open nature, FANETs are particularly vulnerable to various security threats, including wormhole attacks. In a wormhole attack, an attacker creates a tunnel between two malicious nodes, capturing packets at one end and replaying them at the other end almost instantly [32,33]. Wormhole attacks creates the illusion of a shorter and more efficient route between the malicious nodes. When any transmitted packet reaches one of the attacker nodes, that node forwards the packet to its distant malicious counterpart through legitimate intermediate nodes. Although these intermediate nodes are not directly involved in the communication, their resources get drained because of their unwitting participation in the wormhole attack.

In the multicast routing process, each destination node sends a join request to the cluster head (CH) within its cluster. The source node sends a multicast route request (MRREQ) via unicast to each CH to find multiple destination nodes. Upon receiving the MRREQ, the UAV gateway node (UGW) updates its routing table and rebroadcasts the MRREQ to neighboring nodes. When a CH with a relevant multicast group ID receives an MRREQ, it updates its routing table and sends a Route Reply (RREP) packet back to the source node via unicast. Wormhole attackers also forward MRREQ and RREP packets. They exploit this routing process by sending fake RREPs with significantly higher sequence numbers than normal nodes. This makes the attackers appear to offer the freshest route to the destination, even without consulting their routing table. As a result, other nodes update their routing tables based on this misleading information, causing severe disruptions in network functionality.

## 3. The Particle Swarm Optimization Theory

The particle swarm optimization (PSO) algorithm is an evolutionary computation technique developed by Kennedy and Eberhart in 1995 [34]. In this algorithm, a swarm of particles explores a multi-dimensional search space to find optimal solutions. Given an optimization function f(X), where *X* is an *n*-dimensional random vector, these particles serve as candidate solutions for the optimization problem. Each particle *i* is characterized by its velocity $V_{i}=\left(v_{i1},v_{i2},...,v_{ij}\right)$ and position $P_{i}=\left(p_{i1},p_{i2},...,p_{ij}\right)$, i=1,2,...,q, j=1,2,...,n, and *n* and *q* represent the dimensions and swarm size, respectively. Each particle represents a candidate solution and searches for the global optimum in the problem space. To find the optimal solution, each particle moves towards the previous best position (pbest) and the global best position (gbest) in the cluster can be mathematically expressed as [35]
(1)$$\begin{array}{c} \mathrm{pbest}\left(i,t\right)=\arg\min_{k=1,\cdots ,t}\left[f\left(P_{i}\left(k\right)\right)\right],i\in 1,2,\cdots ,N_{p},\\ \mathrm{gbest}\left(t\right)=\arg\min_{\begin{array}{c} i=1,\cdots ,N_{p}\\ k=1,\cdots ,t \end{array}}\left[f\left(P_{i}\left(k\right)\right)\right], \end{array}$$
where $N_{p}$ denotes the total number of particles, *t* denotes the current iteration number, and *f* denotes the fitness function. In each generation, particles *i* adjusts its velocity $V_{i}$ and position $X_{i}$ according to the following formula [36]:(2)$$\begin{array}{c} v_{i}\left(t+1\right)=wv_{i}\left(t\right)+c_{1}r_{1}\left(t\right)\left(\mathrm{pbest}\left(i,t\right)-P_{i}\left(t\right)\right)+c_{2}r_{2}\left(t\right)\left(\mathrm{gbest}\left(t\right)-P_{i}\left(t\right)\right),\\ P_{i}\left(t+1\right)=P_{i}\left(t\right)+v_{i}\left(t+1\right), \end{array}$$
where *w* is the inertia weight used to balance the global exploration and local exploitation, $r_{1}$ and $r_{2}$ are uniformly distributed random variables within range [0,1], and $c_{1}$ and $c_{2}$ are positive constant parameters called acceleration coefficients. As the algorithm progresses, particles share their best-known positions with some or all of the swarm. This collaborative sharing helps the guide of the group toward optimal solutions. The extent to which this division affects individual particles depends on the specific environmental topology used.

## 4. The Proposed Secure Multicast Routing Protocol: DLSMR

In this section, we describe the structural characteristics of our network, the specifications of the UAV and ground node, and the existence and implications of wormhole attacks within the UAV layer.

Our network consists of two types of nodes $K={\left\{F_{k}\right\}}_{k=1}^{K}$ represented by a set of flying nodes F and $L={\left\{G_{l}\right\}}_{l=1}^{L}$ represented by a set of ground nodes G. These nodes are organized into clusters, ${\mathrm{CL}}_{c}≜\left\{c=1,2,\cdots ,C\right\}$, through a top-down methodology that enhances network scalability and performance. UAVs have a distinct set of characteristics that are crucial for our protocol. These flying nodes are mobile and have the ability to cover large ground areas, providing superior network coverage compared to traditional ground nodes. However, their movement speed, direction, and remaining energy must be accurately tracked to maintain network efficiency and stability. In contrast, ground nodes serve a vital purpose as points of connection. They receive, process, and transmit data to UAVs, playing an important role in establishing and maintaining the network’s functionality. One of the major security threats in our setup is the presence of wormhole attacks within the UAV layer. These attacks involve malicious nodes that tunnel packets from one location to another within the network. This deceptive maneuver disrupts regular data transmission pathways and modifies network topology, causing significant disturbances to the network’s performance and reliability. To counteract these challenges, we propose a novel DLSMR protocol and a TD-PSO-based clustering protocol. The DLSMR protocol is designed to predict secure routes and avoid wormhole nodes, thereby securing data transmission. Concurrently, the TD-PSO clustering protocol groups UAVs and ground nodes into clusters based on their positions, speed, direction, and remaining energy, providing stable connectivity and improving overall network performance. Through these techniques, we aim to build a secure, scalable, and efficient communication network that effectively counteracts wormhole attacks while satisfying the unique characteristics and needs of UAVs and ground nodes.

### 4.1. The Basic Concept of the Proposed Secure Multicast Routing and Clustering Protocol

In this section, we explain the basic concepts of our proposed secure multicast routing and clustering protocol. The process of establishing routes is divided into two steps, which are as follows:**Step 1 (Clustering Process):** Firstly, we consider the entire network as a single cluster, which is then recursively divided into smaller clusters based on the weight of the node degree, cosine similarity, cosine distance, and remaining energy. We identify flying nodes as potential cluster heads in this top-down clustering process as they inherently possess stronger capabilities within the cluster. Moreover, the flying node with the highest remaining energy is elected as the CH since it has to work harder than the member nodes as noted in [23]. A flying node that is not selected as a cluster head receives two or more cluster head information (CHI) packets from the CH, and then that node becomes an UGW. Otherwise, the ground nodes can only serve as cluster members. To select the optimal cluster head, the cluster member candidates use the PSO algorithm to optimize costs. Unlike the traditional clustering method of [23], we introduce a TD-PSO protocol to improve control overhead during cluster formation.**Step 2 (Multicast Routing Process):** Following the clustering process, each destination node that wants to receive data transmission sends a join request to the CH within the cluster. Then, a source node sends a MRREQ to each cluster head by unicast to find the multiple destination nodes. After that, CH will broadcast the MRREQ packet to each CH to find the multiple destination nodes. When the UGW receives the MRREQ packet, it updates its routing table and re-broadcasts the MRREQ packet to the neighbor nodes. When a CH who has a multicast group ID receives an MRREQ packet, they update the routing table and reply a RREP packet to the previous node by unicast. In addition to that, the wormhole attacker also forwards the MRREQ and RREP packets. In this work, we develop a novel DL framework to predict the secure next node while establishing the route from a source to the multicast group destinations. Based on the DL framework, we can train a DNN model to learn wormhole nodes’ characteristics and distinguish them from legitimate ones. Each flying node in the network utilizes a deep learning framework to find the next secure nodes for multiple destinations. We consider the node ID, node’s position, destination ID, destination sequence, hop count, and the remaining energy of the node as the trainable input parameter, and the secure and unsecured node ID, secure and unsecured status as output parameters of the DNN model. When a node receives a RREP packet, the contents of the relevant RREP packet will be used in the DL framework. After training, the DNN model produces secure and unsecured node ID and their status as output. For example, as we can see in Figure 1, when the CH receives two RREP packets which are from the neighbor nodes and the wormhole nodes, the proposed DLSMR protocol can establish a secure multicast route from $S-{\mathrm{CH}}_{1}-{\mathrm{UGW}}_{\left(1,4\right)},{\mathrm{CH}}_{2}$ and also $S-{\mathrm{CH}}_{1}-{\mathrm{UGW}}_{\left(1,4\right)},{\mathrm{CH}}_{2},{\mathrm{CH}}_{3}$ with a multicast tree to avoid the wormhole attack.

### 4.2. The Proposed Clustering Protocol: Top-Down Particle Swarm Optimization (TD-PSO)

#### 4.2.1. The Basic Concept of the TD-PSO

As shown in Figure 2, we design the top-down particle swarm optimization-based clustering protocol which ensures network connectivity and reduces the control overhead in FANETs. The proposed clustering protocol considers the problem of the join weight of the node degree, cosine similarity, cosine distance, and remaining energy to form clusters. Additionally, we use the higher remaining energy among the candidate nodes to select the cluster head. In this work, we assume only UAV can become the CH because it has the greatest resources compared to the ground node. Then, the selected CHs transmit packets through inter-cluster forwarding. In most cases, this top-down approach can ensure network connectivity and coverage. Unlike the bottom-up method, the top-down method can reduce the number of control overhead because the number of flying nodes (FN) is greater than the number of ground node (GN) nodes with CF-mMIMO properties. In order to simplify, we will refer to UAVs and ground users as nodes. The proposed TD-PSO protocol assumes that each node can know its location information by using the global positioning system (GPS). The following subsubsection will explain the proposed TD-PSO protocol in detail.

#### 4.2.2. The Proposed Clustering Protocol: TD-PSO

The proposed TD-PSO protocol considers node position, node speed, node direction, and remaining energy to form the clusters and elect the cluster head. Figure 3 illustrates the flowchart of the proposed TD-PSO clustering protocol.

The procedure of forming a cluster and electing the cluster head can be explained as follows:**Step 0: Initialization**The nodes turn on and operate independently when the simulation starts. Then, go to **step 1**.**Step 1: Dissemination of Node Information**Each flying node $f_{k}$ periodically estimates its information such as speed, position, direction, and remaining energy. Then, $f_{k}$ generates and broadcasts information (INFO) packets to its neighbor UAV nodes. The INFO packet contains the following fields:
$$\langle \mathrm{Type},S_{\mathrm{ID}},D_{\mathrm{ID}},E,U_{\mathrm{type}}\rangle$$
where Type represents packet type, $S_{\mathrm{ID}}$ represents source node ID, $D_{\mathrm{ID}}$ represents destination node ID, E represents the remaining energy of each node, and  $U_{\mathrm{type}}$ represents the type of node (0 is G, 1 is F). Then, go to **step 2**.-If ground node $g_{l}$ receives an INFO packet, then it drops the packet; go to **step 4**. Otherwise, go to **step 4**.**Step 2: Election of Cluster Head**In order to support strong connectivity, the cluster head is selected by the most remaining energy among the candidate flying nodes, which mathematically can be expressed as
(3)$$f_{k}^{*}=\arg\max_{f_{k}\in {\mathrm{NB}}_{f_{k}}^{*}\cup \left\{f_{k}\right\}}\left\{E_{k}\right\}.$$-If $f_{k}$ = $f_{k}^{*}$, the node $f_{k}$ becomes the cluster head; go to **step 3**.-Otherwise, wait to receive the CHI packet and go to **step 4**.**Step 3: Dissemination of Cluster Head Information**The node $f_{k}$ becomes the cluster head, then generates and broadcasts the CHI packet to be announced to its neighbor nodes. The CHI packet contains the following fields:
$$\langle \mathrm{Type},S_{\mathrm{ID}},D_{\mathrm{ID}},\mathrm{Pos},\mathrm{Dir},S,E,U_{\mathrm{type}}\rangle$$
where Type represents packet type, $S_{\mathrm{ID}}$ represents the source node ID, $D_{\mathrm{ID}}$ represents the destination node ID, Pos represents the node’s position, Dir represents the direction of the node, S represents the speed of the node, E represents the remaining energy of the node, and $U_{\mathrm{type}}$ represents the type of the node (0 is G, 1 is F).-If $f_{k^{{}^{\prime }}}\in F\setminus \left\{f_{k}\right\}$ receive two or more CHI packets, then go to **step 5**.-Otherwise, go to **step 4**.**Step 4: Decision of Member Node**When ground node $g_{l}$ receives two or more CHI packets, $g_{l}$ will decide which cluster head follows by calculating the cost function of the cluster head candidate. The ground node will select the cluster head candidate with the largest cost value. The objective is to maximize the cost value by considering weight value under node degree, cosine similarity, cosine distance, and energy, which can be formulated as
(4a)$$\begin{array}{cc} \;\;\;\;\;\max_{\left\{x_{1},x_{2},x_{3},x_{4}\right\}} & C_{l}=x_{1}\Delta_{l}+x_{2}{\mathrm{Cosim}}_{l}+x_{3}{\mathrm{Dist}}_{l}+x_{4}E_{l}, \end{array}$$
(4b)$$\begin{array}{cc} \;\;\;s.t\; & x_{1}+x_{2}+x_{3}+x_{4}=1, \end{array}$$
(4c)$$\begin{array}{c} \Delta_{l}\leq \Delta_{\mathrm{th}}, \end{array}$$
(4d)$$\begin{array}{cc} \; & {\mathrm{Cosim}}_{l}\geq {\mathrm{Cosim}}_{\mathrm{th}}, \end{array}$$
(4e)$$\begin{array}{cc} \; & {\mathrm{Dist}}_{l}\leq {\mathrm{Dist}}_{\mathrm{th}}, \end{array}$$
(4f)$$\begin{array}{c} E_{l}\geq E_{\mathrm{th}}, \end{array}$$
where (4b) denotes that the total weight of the particle must be equal to one, (4c) denotes that the node degree difference must be lower than or equal to the node degree threshold, (4d) denotes that the cosine similarity between two nodes must be greater than or equal to the cosine similarity threshold, (4e) denotes that the cosine distance between nodes must be lower than or equal to the cosine distance threshold, and (4f) denotes that the energy of the cluster head must be greater than or equal to the energy threshold. We consider four factors which consist of the node degree difference, cosine similarity, the cosine distance between two nodes, and the remaining energy of its node. The node degree of its nodes can be written as
(5)$$D_{i}=\sum_{j=1,j\neq i}^{n}H_{i}^{j}$$
(6)$$H_{i}^{j}=\sum_{i=0}^{n}\left\{\mathrm{dist}\left(i,j\right)\right\}<R_{i}\}$$
where dist(i,j) can be defined as
(7)$$\mathrm{dist}\left(i,j\right)=\sqrt{{\left(x_{i}-x_{j}\right)}^{2}+{\left(y_{i}-y_{j}\right)}^{2}+{\left(z_{i}-z_{j}\right)}^{2}}.$$$D_{i}$ stands for the node degree of the node $f_{i}$, $R_{i}$ stands for the communication range of the node *i*, and dist(i,j) stands for distance between node *i* and *j*. Then, the average node degree can be expressed as
(8)$${\bar{D}}_{i}=\frac{\sum_{i=1}^{n}D_{i}}{n}.$$The self-adaptive node degree variance is calculated by subtracting the node measure from its average measure, which can be expressed as:
(9)$$\Delta_{i}=\left|\mathrm{dist}\left(i,j\right){\bar{D}}_{i}\right|$$The second factor is cosine similarity between two nodes which can be defined as [29,37]
(10)$$\mathrm{CoSim}\left(i,j\right)=\frac{\sum_{n_{i}=1}^{n_{I}}{\vec{V}}_{n_{i}}{\vec{V}}_{j}}{\sqrt{\sum_{i=1}^{N}{\vec{V}}_{i}^{2}}\sqrt{\sum_{j=1,m\neq i}^{N}{\vec{V}}_{j}^{2}}},$$
where ${\vec{V}}_{i}$ and ${\vec{V}}_{j}$ are the *i*-th and *j*-th nodes’ vector information, respectively. Each node ${\vec{V}}_{i}$ is related with a mobility vector information metric value (i.e., speed, direction, and position) ${\vec{V}}_{i}=({\vec{V}}_{1}$, ${\vec{V}}_{2}$,⋯,${\vec{V}}_{j})$, where ${\vec{V}}_{i}$ constitutes the vector values which indicate link information between nodes. Under a constrained communication distance, we can control the cluster member by considering the maximum cosine similarity. Afterwards, the third factor is the cosine distance of the node, which is used to find the distance between two nodes and can be formulated by [38]
(11)CoDis(i,j)={1−CoSim(i,j)}.Thus, the selected cluster member can be mathematically formulated as
(12)$$j^{*}=\arg\max_{j}\left\{C_{i},C_{j}\right\}$$
where $C_{i}$ represents the cost of the node *i* and $C_{j}$ represents the the cost of the neighbor cluster heads near node *j*. After that, $g_{l}$ sends the join cluster (JC) packet to the CH. The JC packet contains the following fields:
$$\langle \mathrm{Type},S_{\mathrm{ID}},D_{\mathrm{ID}},U_{\mathrm{type}}\rangle$$
where Type represents packet type, $S_{\mathrm{ID}}$ represents source node ID, $D_{\mathrm{ID}}$ represents destination node ID, and $U_{\mathrm{type}}$ represents the type of node. Then, go to **step 5**.-Otherwise, when $g_{l}$ only receives one CHI packet, then sends the JC packet to the CH and go to **step 6**.**Step 5: Decision of Gateway Node**The flying node $f_{k^{{}^{\prime }}}\in F\setminus \left\{f_{k}\right\}$ receives two or more CHI packets and become a gateway node. Similar to the ground nodes, the flying node $f_{k^{{}^{\prime }}}$ will decide which cluster head follows by calculating the cost function of the cluster head candidate. When the value of the cost function is larger, the relationship between the cluster head and node is better than other relationships which can be mathematically expressed as
(13a)$$\begin{array}{cc} \;\;\;\;\max_{\left\{x_{1},x_{2},x_{3},x_{4}\right\}}\; & C_{k}=x_{1}\Delta_{k}+x_{2}{\mathrm{Cosim}}_{k}+x_{3}{\mathrm{Dist}}_{k}+x_{4}E_{k}, \end{array}$$
(13b)$$\begin{array}{cc} \;\;\;s.t\; & x_{1}+x_{2}+x_{3}+x_{4}=1, \end{array}$$
(13c)$$\begin{array}{c} \Delta_{k}\leq \Delta_{\mathrm{th}}, \end{array}$$
(13d)$$\begin{array}{cc} \; & {\mathrm{Cosim}}_{k}\geq {\mathrm{Cosim}}_{\mathrm{th}}, \end{array}$$
(13e)$$\begin{array}{cc} \; & {\mathrm{Dist}}_{k}\leq {\mathrm{Dist}}_{\mathrm{th}}, \end{array}$$
(13f)$$\begin{array}{c} E_{k}\geq E_{\mathrm{th}}, \end{array}$$
where (13b) denotes that the total weight of the particle must be equal to one, (13c) that denotes the node degree difference must be lower than or equal to the node degree threshold, (13d) denotes that the cosine similarity between two nodes must be greater than or equal to the cosine similarity threshold, (13e) denotes that the cosine distance between nodes must be lower than or equal to the cosine distance threshold, and (13f) denotes that the energy of the cluster head must be greater than or equal to the energy threshold. Thus, the selected cluster member can be mathematically formulated as
(14)$$j^{*}=\arg\max_{j}\left\{C_{k},C_{j}\right\}$$
where $C_{k}$ represents the cost of the node *k* and $C_{j}$ represents the cost of the neighbor cluster heads near node *j*. Then, $f_{k^{{}^{\prime }}}$ sends the JC packet to the CH; go to **step 6**.-Otherwise, when $f_{k^{{}^{\prime }}}$ only receives one CHI packet, then it sends the JC packet to the CH; go to **step 6**.**Step 6: Cluster Member Table Updates**When node $f_{k}^{{}^{\prime }}$ and $g_{l}$ choose the CH, they update the cluster member (CM) table as shown in Table 1 where ${\mathrm{CM}}_{\mathrm{ID}}$ is a unique identifier assigned to each node within the cluster, ${\mathrm{CH}}_{\mathrm{ID}}$ represents the identifier of the cluster head that the cluster member belongs to, and $U_{\mathrm{type}}$ represents the type of user. Then, go to **step 7**.**Step 7: Cluster Head Table Updates**When $f_{k}$ receives a JC packet, $f_{k}$ replies with an accept packet (AC) to the transmitted node, and updates the cluster head table as shown in Table 2 where ${\mathrm{CH}}_{\mathrm{ID}}$ is a unique identifier assigned to each cluster head in the network, ${\mathrm{CM}}_{\mathrm{ID}}$ represents the identifier of the cluster member that belongs to the cluster managed by the cluster head, and $U_{\mathrm{type}}$ represents the type of user. Finally, the cluster has been formed. The AC packet contains the following fields:
$$\langle \mathrm{Type},S_{\mathrm{ID}},D_{\mathrm{ID}}\rangle$$
where $\mathrm{Type},S_{\mathrm{ID}},\mathrm{and}\;D_{\mathrm{ID}}$ represent the packet type, source node ID, and destination node ID, respectively.-Otherwise, the **AC** packet will be dropped.

In the end, the clustering process is finished. When the source node needs to send a data transmission to multiple destination nodes, the network will start the routing process based on the routing protocol algorithm. The list of packets for the proposed TD-PSO clustering protocol is summarized in Table 3.

#### 4.2.3. Top-Down Particle Swarm Optimization Model

This subsection will explain in detail the particle swarm optimization model used in the clustering protocol.

In the TD-PSO algorithm, each individual node in the population is called a particle and moves in the search space. Particles have memory and, thus, they retain part of their previous state. Each particle’s movement is the composition of a velocity and two randomly weighted influences. The two randomly weighted influences are individual and have the tendency to return to their best previous positions and sociality, or the tendency to move towards their neighborhood’s best previous position. As briefly mentioned above, clustering involves gathering similar objects that must first be defined.

Let us assume the objective function $C_{i}$ where i∈{FN,GN} is the global optima $C_{i}^{*}$ of optimization with optimal value $x_{l}$, l∈{1,2,3,4}. The proposed TD-PSO algorithm to find the optimal values, i.e., $x_{1},x_{2},x_{3},x_{4}$, can be summarized as Algorithm 1, where *p* represents the number of parameters, *q* denotes the number of particles, $x_{pq}$ represents the position of particle *q* for parameter *p*, $v_{pq}$ represents the velocity of particle *q* for parameter *p*, $r_{1}$ and $r_{2}$ are random values with range [0, 1] to avoid premature convergence, *w* denotes the weight of a particle with range [0.4, 0.9], $c_{1}$ and $c_{2}$ are acceleration factors, and *t* is the number of iterations.
**Algorithm 1** The TD-PSO algorithm to find global optimal points in problems (4) and (13)**Output:** Optimal solution $C^{*}$ and $x_{l}^{*}$
  1:*Initialization* :  2:Set *p*← 4. ← number of parameters  3:Set *q*← 50. ← number of particles  4:Set ($x_{pq},v_{pq}$) ← randomly with constraint.  5:Set *w*← randomly with range [0.4,0.9].  6:Set ($r_{1},r_{2}$) ← randomly with range [0,1].  7:Set ($c_{1},c_{2}$) ← 2.  8:**for** κ=0,1,2,⋯**do**  9:    calculate problem, find $C^{*},x_{l}^{*}$;10:    **if** $C^{*}<C^{\left(t\right)}$ **then**11:        $C^{*}$←$C^{\left(t\right)}$12:        $x_{p}^{*}$←$x_{p}^{\left(t\right)}$13:    **end if**14:    $v_{pq}^{\left(t+1\right)}=w.v_{pq}^{\left(t\right)}+c_{1}.r_{1}\left(x_{bl}^{\left(t\right)}-x_{pq}^{\left(\kappa \right)}\right)+c_{1}.r_{1}\left(g_{bl}^{t}-x_{pq}^{\left(t\right)}\right)$15:    $x_{pq}^{\left(t+1\right)}=x_{pq}^{\left(t\right)}+v_{lj}^{\left(t\right)}$16:**end for**


### 4.3. The Proposed Deep Learning-Based Secure Multicast Routing Protocol: DLSMR

We describe the proposed DLSMR protocol to establish a high-stability multicast tree and improve security performance as shown in Figure 4. The objective function of this approach is to predict the secure and unsecured nodes during the routing process. Due to the node’s mobility changing rapidly in FANETs, it makes it easy to attack the network. In particular, wormhole attacks pose a high risk when the nodes can manipulate packets during routing and data transmission. Furthermore, also, even when we consider unicast transmission, it makes the control overhead very high. Thus, this paper proposes deep learning-based secure multicast routing to avoid wormhole attacks and improve control overhead in FANETs. It differs from previous works for solving wormhole attacks using bio-inspired, position-based, and distance methods. In this approach, we consider deep learning to predict the secure and unsecured nodes quickly during the routing process. The use of this process can mitigate these threats, ensuring reliable and secure communication between nodes. Additionally, multicast routing protocols can accommodate this dynamic nature by allowing for frequent group membership changes and route updates. As can be observed in Figure 4, a source node (S) needs to establish a multicast tree to the multiple destination nodes (D). After the clustering process is completed, each D that wants to receive data transmission sends a join request to the CH in the cluster. Then, S sends a MRREQ packet to each CH by unicast to find the multiple destination nodes. After that, CH will broadcast the MRREQ packet to each CH to find the multiple destination nodes. When UGW receives the MRREQ packet, it updates its routing table and re-broadcasts the MRREQ packet to the neighbor nodes. When a CH who has a multicast group ID receives an MRREQ packet, it updates the routing table and replies a RREP packet to the previous node by unicast. We design the DL framework with the node ID, node position, destination ID, destination sequence, hop count, and remaining energy as input parameters, and secure and unsecured node ID, secure and unsecured status as output parameters. By using a DL framework, we can determine whether the next node is secure or not to establish a secure multicast route from S to multiple destinations as detailed in Section 4.4.

Figure 5 illustrates the flowchart of the proposed secure multicast routing protocol, which can be summarized as follows:


**Route Request Process:**

**Step 1: Initialization**
After the clustering process is completed, each destination node (multicast member node) in a cluster that wants to receive certain data sends a join request (JREQ) to its cluster head by unicast. The JREQ contains the following:
$$\langle \mathrm{Type},S_{\mathrm{ID}}^{\mathrm{JREQ}},D_{\mathrm{ID}},\mathrm{Status}\rangle$$
where Type denotes packet type, $S_{\mathrm{ID}}^{\mathrm{JREQ}}$ identifies the node that wants to join a multicast group, $D_{\mathrm{ID}}$ denotes the node ID that wishes to join, and Status denotes the status of the node (join, leave, etc.). Next, the cluster head stores the multicast ID ($M_{\mathrm{ID}}$) associated with this request in its own table; go to **step 2**.
**Step 2: Source Node Operation for Route Request: Generates and Sends Route Request Packet**
A node that wants to send data to the multicast group becomes the S. S initiates the process by generating a MRREQ packet and unicasts this MRREQ to its CH. The MRREQ packet contains the following fields:
$$\langle \mathrm{Type},S_{\mathrm{ID}}^{\mathrm{MRREQ}},{\mathrm{MG}}_{\mathrm{ID}},M_{\mathrm{ID}},S_{\mathrm{Seq}},\mathrm{Hop},\mathrm{TTL}\rangle$$
where Type represents packet type, $S_{\mathrm{ID}}^{\mathrm{MRREQ}}$ represents the source node ID, ${\mathrm{MG}}_{\mathrm{ID}}$ represents the multicast group ID, $M_{\mathrm{ID}}$ represents the multicast ID, $S_{\mathrm{Seq}}$ is the source sequence number, Hop denotes the number of hops between two nodes, and TTL is the time to live of the packet in the network. Otherwise, go to **step 3**.
**Step 3: Intermediate Node Operation at Cluster Head for Route Request**
When ${\mathrm{CH}}_{c}$ receives the MRREQ packet, ${\mathrm{CH}}_{c}$ will first check the MRREQ packet.
▪
**Step 3.1:** If the $S_{\mathrm{Seq}}$ at the received MRREQ is larger than that of the routing table, then go to **step 3.2**.*Conversely, if the $S_{\mathrm{Seq}}$ at the received MRREQ equals the $S_{\mathrm{Seq}}$ at the routing table and the $M_{\mathrm{ID}}$ at the received MRREQ is equal to the $M_{\mathrm{ID}}$ at the routing table, or the $M_{\mathrm{ID}}$ is greater than the $M_{\mathrm{ID}}$ at the routing table and {hop+1} at the received MRREQ is less than hop at the routing table, then go to **step 3.2**. Otherwise, the packet will be dropped.
▪
**Step 3.2:** If the TTL at the received MRREQ is greater than or equal to the TTL at the routing table, then go to **step 3.3**. Otherwise, the packet will be dropped.
▪
**Step 3.3:** If the cluster member ID (${\mathrm{CM}}_{\mathrm{ID}}$) is the same as the ${\mathrm{MG}}_{\mathrm{ID}}$, ${\mathrm{CH}}_{c}$ records the sender’s ID, updates the routing table, and broadcasts MRREQ to ${\mathrm{NB}}_{i}$; then, go to **step 5**.*Otherwise, ${\mathrm{CH}}_{c}$ records the sender’s ID and updates the routing table and broadcasts MRREQ to the neighbor node (${\mathrm{NB}}_{i}$) in its cluster or the next cluster heads; go to **step 4**.
**Step 4: Intermediate Node Operation at Gateway for Route Request**
When the gateway node ${\mathrm{UGW}}_{c}$ receives a MRREQ packet from ${\mathrm{NB}}_{i}$, ${\mathrm{UGW}}_{c}$ records the sender’s ID and updates the routing table, then ${\mathrm{UGW}}_{c}$ broadcasts the MRREQ to their neighboring CH nodes until $\mathrm{TTL}\geq {\mathrm{TTL}}^{\mathrm{th}}$; go to **step 3**. 



**Route Reply Process:**

**Step 5:**
**Cluster head Operation for Route Reply: Generates and Sends Route Reply Packet**
CH generates and replies a RREP packet to the previous node by unicast transmission. The RREP packet contains the following fields:
$$\langle \mathrm{Type},S_{\mathrm{ID}}^{\mathrm{RREP}},D_{\mathrm{ID}}^{\mathrm{RREP}},M_{\mathrm{ID}},D_{\mathrm{Seq}},E,\mathrm{Pos},\mathrm{Hop}\rangle$$
where Type represents the packet type, $S_{\mathrm{ID}}$ represents the source node ID, $D_{\mathrm{ID}}$ represents the destination ID, $M_{\mathrm{ID}}$ represents the multicast ID, $D_{\mathrm{Seq}}$ is the destination sequence number which is the number of attempts to confirm control messages, E represents the remaining energy of the node, Pos represents the position of the node, and Hop represents the number of hops to D. Then, go to **step 6**.
**Step 6: Intermediate Node Operation at Gateway Node for Route Reply**
The intermediate node ${\mathrm{NB}}_{i}$ records the sender’s ID of RREP packets and updates its routing table when it receives the RREP packet. Then, ${\mathrm{NB}}_{i}$ forwards the RREP packet to the previous node; go to **step 7**. Otherwise, ${\mathrm{NB}}_{i}$ waits until it receives the RREP packet.
**Step 7: Intermediate Node Operation at CH for Route Reply**
When the ${\mathrm{CH}}_{c}$ receives RREP, then go to **step 7.1**. Otherwise, ${\mathrm{CH}}_{c}$ waits until it receives the RREP packet.
▪
**Step 7.1:** When the ${\mathrm{CH}}_{c}$ receives the RREP, then ${\mathrm{CH}}_{c}$ checks the $S_{\mathrm{ID}}^{\mathrm{RREP}}$ in the RREP packet. The $S_{\mathrm{ID}}^{\mathrm{RREP}}$ in the RREP packet means the destination node ID. So, if the $S_{\mathrm{ID}}^{\mathrm{RREP}}$ equals $S_{\mathrm{ID}}^{\mathrm{tab}}$, then go to step **step 7.2**. Otherwise, ${\mathrm{CH}}_{c}$ records the sender’s ID, updates the routing table, and forwards to the previous node by unicast transmission; then, go to **step 8**.
▪
**Step 7.2:** When the $S_{\mathrm{ID}}^{\mathrm{RREP}}$ equals $S_{\mathrm{ID}}^{\mathrm{tab}}$, we will predict the best secure next node with input parameter consisting of the node ID, position of the node, destination ID, destination sequence number, and remaining energy. The output parameters consists of the ${\mathrm{NID}}_{\sec}^{1}$, ${\mathrm{NID}}_{\mathrm{nsec}}^{2}$, and the status of the node (secure or unsecured) to select the secure next node as the best next node while establishing the route from S to multiple $D_{i}$ by using a DNN model; then, go to step **step 7.3**.
▪
**Step 7.3:** If the ${\mathrm{Stat}}_{\sec}$ equals 1, then ${\mathrm{CH}}_{c}$ can determine the secure route to be pursued by the secure next node. Then, ${\mathrm{CH}}_{c}$ records the ${\mathrm{NID}}_{\sec}$, updates the routing table, and forwards to the previous node. The multicast tree routing table can be summarized in Table 4, where $S_{\mathrm{ID}}$ is the source ID, ${\mathrm{MG}}_{\mathrm{ID}}$ is the multicast group ID, $S_{\mathrm{Seq}}$ is the source sequence number, $D_{\mathrm{Seq}}$ is the destination sequence number , $M_{\mathrm{ID}}$ is the multicast node ID, ${\mathrm{PN}}_{\mathrm{ID}}$ is the previous node ID, ${\mathrm{NN}}_{\mathrm{ID}}$ is the next node ID, and Hop represents the number of hops to D. The routing table will be used to determine the next node to the multicast group D that data packets will pass through during the data transmission process. Then, go to **step 8**. Otherwise, the packet will be dropped and the process is ended.
**Step 8: Source Node Operation for Route Reply**
If S receives all RREP packets from the multicast group, then go to the data transmission process in **step 9**. Otherwise, go to step **Step 8.1:**.
▪
**Step 8.1:** If $\mathrm{Timer}\leq 2\times {\mathrm{TTL}}^{\mathrm{th}}$, wait for all RREP packets from the multicast group until $\mathrm{Timer}>2\times {\mathrm{TTL}}^{\mathrm{th}}$. Otherwise, go back to **step 2**.



**Data Transmission Process:**

**Step 9: Data Transmission at Source Node**
When S receives all RREP packets from the multicast group $D_{i}$, the S multicasts the data packet to the next hops based on the deep learning framework. If a node of the tree receives a data packet, it will forward the data packet to the multicast group in the same way as the source.


The list of packets for the DLSMR protocol are summarized in Table 5.

### 4.4. The Proposed Deep Learning Design

As shown in Figure 1, a wormhole attack occurs when an attacker creates a tunnel between two malicious nodes in the network, allowing them to capture packets at one end and replay them at the other end instantly. A wormhole creates the illusion of a shorter and more efficient route between the two malicious nodes. To solve this problem, we develop a novel deep learning framework to predict the secure next node while establishing the route from the source to multiple destinations in FANETs.

Each node in the network can determine which node in the network is the most secure for multiple destinations through a deep learning framework. In this subsection, we design the deep learning framework to capture the relation between network parameters and system performances as shown in Figure 6. The main objective of this work is to predict the secure next node in the proposed DLSMR protocol. In this work, we utilize multivariate regression, which is more challenging than single regression. A deep learning model includes two phases, the training phase and the testing phase.

In the training phase, the input parameters consist of the node identifier (NID), the distance between two nodes (Dist) which are denoted in Equation (Equation 7), the destination sequence ($D_{\mathrm{seq}}$), the number of hops (Hop), and the remaining energy of the node (E). According to these input parameters, the model predicts the secure node ID, the unsecured node ID, the secure status, and the unsecured status as the output.

In a training iteration, as shown in Figure 6a, an error is obtained by comparing the deep learning output with the target and the simulation result obtain four outputs. Then, the error is minimized by updating the weights and biases on the neurons using back-propagation, which continues until the iteration is satisfied. The trainable deep learning framework is tested using a new input variable to predict the secure and unsecured next nodes as shown in Figure 6b. In our design, we use a feed-forward neural network with 1×5 dimensional input layers, *L* hidden layers, and 1×4 dimensional output layers to obtain the two kinds of ID and status of the node as shown in Figure 7.

Therefore, the proposed deep learning design can reduce complexity and also can predict the secure next node in real-time based on the information of its node as the input parameter. The layer structure used in the deep learning design to improve system performance is shown in Table 6.

## 5. Performance Evaluation

### 5.1. Simulation Environments and Parameters

In this section, we evaluate the performance of the proposed routing protocol, DLSMR. To illustrate our proposed protocol, we deploy 30, 50, and 100 nodes within an urban area of 1000 m × 1000 m and a transmission range of 250 m. In addition to that, we deploy a wormhole pair consisting of two wormhole nodes. These wormhole nodes are placed randomly within the network for each simulation run. We establish a ‘tunnel’ between these randomly positioned wormhole nodes to simulate the wormhole attack, allowing them to shortcut the normal network routing. In this scenario, we simulate our proposed protocol under two different mobility models, namely random waypoint mobility (RWP) [39] and reference point group mobility (RPGM) [40], to evaluate its performance in FANETs. Every result in this simulation is an average from 200 sections, with pausing and moving times set at 3 s and 5 s, respectively. Furthermore, the mobile nodes are initially randomly distributed around the simulated area and move at different speeds (15 km/h, 30 km/h, 45 km/h, 60 km/h, and 75 km/h). The MAC protocol is modeled using the IEEE 802.11a standard and uses a receiver signal strength indicator (RSSI) threshold of −80 dbm for communication range to make it more practical. The main reason for considering RSSI is that the value of RSSI fluctuations obtained has taken into consideration its effect on changes in channel conditions, including multi-path fading [41].

The simulation experiments are conducted using the NS3 simulator. We summarize the simulation environment and parameters in Table 7. Additionally, we measured the accuracy between the predicted secure next node and the output data of the test set by calculating the root mean square error (RMSE) of the proposed deep learning framework. The RMSE can be written as [42]
(15)$$\mathrm{RMSE}=\sqrt{\frac{1}{n}\sum_{t=1}^{n}{\left(p_{\left(t\right)}-{\hat{p}}_{\left(t\right)}\right)}^{2}}$$
where *n* denotes the number of samples in the test set, $p_{\left(t\right)}$ denotes the predicted value of the *t*-th observation in the dataset, and ${\hat{p}}_{\left(t\right)}$ denotes the observed value for the *t*-th observation in the dataset. It is more likely that the predicted secure next node and observation are closely matched when the RMSE is smaller.

### 5.2. Performance Metrics

The performance metrics used in this paper for performance evaluation purposes are the following [29]:Packet delivery ratio (PDR): This is defined as the number of data packets delivered to multicast destinations over the number of data packets supposed to be delivered to multicast destinations. This ratio represents the effectiveness of the routing strategy.Control overhead: This refers to the average number of control packets sent to nodes during the route creation process per session per node per multicast data delivered.Delay: This is defined as the average delay to establish a multicast route from source to multicast destinations per one session.The average number of cluster head changing: This refers to the number of cluster heads changing per cluster per session on average.Packet loss ratio: This is defined by the proportion of data packets that are lost during transmission from the sender in a multicast group. A lower packet loss ratio indicates better performance and reliability of the secure multicast routing protocol.

### 5.3. Numerical Results

In this subsection, we present numerical results to validate the efficacy of our proposed DLSMR protocol. The simulation settings are outlined in Table 7. We use the NS3 simulation, where the algorithm is run for 200 s with 5 s for each session. For the DNN model, the dataset is generated over 1,000,000, 90% of which are used for training while the remaining 10% are allocated for validation. Additionally, we construct 100 distinct datasets to evaluate the performance of the trained DNN model. Our objective is to predict the secure next node through the DNN model accurately. The parameters employed for DNN training are detailed in Table 8.

To demonstrate the effectiveness of the proposed algorithm (DLSMR with TD-PSO), we compare its performance with the multicast ad hoc on-demand distance vector (MAODV) routing protocol with or without TD-PSO clustering protocol.

First, we evaluate the impact of the number of iterations on the maximum cost of the TD-PSO clustering algorithm in Figure 8. As can be observed in Figure 8, the proposed Algorithm 1 converges at the fifth iteration with the number of population (nPop) of 100, which shows the proposed TD-PSO for the clustering protocol is efficient to find the optimal value. The reason is that the algorithm has a higher chance of generating a solution close to the global optimum during the initial stages. Moreover, the algorithm with a population of 100 outperforms the algorithm with a population of 50. The reason is that when the population size increases from 50 to 100, the algorithm has a larger set of candidate solutions to find the optimal solution.

Figure 9 shows the comparison of the average number of cluster head changes in each session as a function of node speed. As can be seen in Figure 9, when the number of nodes increases from 30 to 100, the average number of cluster head changes increases. Furthermore, when node speed increases from 15 to 75 km/h, the average number of cluster head changes also increases. The reason is that, when the node’s speed increases, the node’s relative position can change quickly. Therefore, the nodes’ arrangement and connections can also be changed rapidly. In the clustering protocol, if nodes are moving quickly, the optimal choice for the cluster head can change frequently as nodes move in and out of range of each other. This might require frequent re-election of the cluster head, increasing the average number of cluster head changes. Additionally, higher speeds might lead to increased link breakages, which would necessitate the formation of new clusters, further increasing the frequency of cluster head changes. Furthermore, when comparing the mobility models, the proposed protocol with the RPGM model outperforms that with the RWP model. In RPGM, nodes in a group follow a predefined reference point, resulting in more stable network topology and less frequent cluster head changes. Additionally, RPGM also captures the correlation of movement between nodes in a group, while in the RWP model each node moves independently, leading to more frequent changes in network topology and causing more cluster head changes. Therefore, in the proposed clustering protocol in a scenario where the number of nodes (30, 50, 100) and their speed increases, the average number of cluster head changes increases slightly when the network topology changes rapidly. Nevertheless, the number of cluster head changes in each session is less than one. This means the clustering protocol has a very high level of stability.

In Figure 10, we analyze the impact of the number of hidden neurons on the DNN model with the different numbers of hidden layers. As we can see in Figure 10, the RMSE with one hidden layer will decrease from 0.3164 to 0.0138, with two hidden layers will decrease from 0.0583 to 0.006, and with five hidden layers will decrease from 0.036 to 0.000124 when the number of hidden neurons increases from 5 to 250. It can be explained that the DNN model with more neurons performs better than the DNN model with fewer neurons. Furthermore, the more hidden layer of the DNN model performs better than the less hidden layer.

Figure 11 illustrates the PDR versus the variation in node speed. As can be observed in Figure 11, when the speed of the node increases, the PDR is decreased. The reason is that the entire network becomes more unstable as node speed increases. This instability leads to more frequent disruptions of the multicast tree structure and potential packet losses, decreasing the PDR. Again, when comparing the two mobility models, it is evident that the RPGM protocol results in superior performance compared to the RWP model, which exhibits a high PDR. However, it should be noted that the reduction in PDR is significantly less in the case of the DLSMR+TD-PSO protocol compared to other schemes. Therefore, the DLSMR+TD-PSO protocol with the RPGM model shows the highest level of reliability in maintaining a high PDR.

Figure 12 reveals the impact of node speed on routing delays, including the latency time for cluster construction per session as a function of node speed. As can be observed in Figure 12, the routing delay increases when the node speed increases. The reason is that nodes will move more dynamically when the node’s speed increases, consequently increasing the time to establish the route. Once again, the protocols under the RPGM model demonstrate higher stability than protocols under the RWP models because the coordinated group movement in RPGM reduces sudden changes in network topology, thereby reducing route establishment delays. On the other hand, the proposed DLSMR+TD-PSO protocol only involves UAVs as CHs and GW nodes to determine the route to be followed by packets. Thus, the proposed DLSMR+TD-PSO protocol with the RPGM model can send packets with a minimum delay compared to other schemes.

Figure 13 presents a comparison of the control overhead as a function of node speed, including the control overhead for cluster formation per node per session. As can be seen, the control overhead increases a little bit when the speed of the node increases. Essentially, this is due to the fact that, when the node’s speed increases, more packets will be needed for route establishment and thus the overhead will rise. Again, comparing the different mobility models, the RPGM model demonstrates better control overhead due to high stability than the RWP model. In addition to that, the proposed TD-PSO clustering protocol can reduce the control overhead in all schemes. This means that the TD-PSO clustering protocol only involves CH and GW in the routing process, and the control overhead decrease is much less in the scheme with TD-PSO clustering protocol compared with schemes without TD-PSO clustering protocol. On the other hand, the proposed DLSMR routing protocol outperforms the MAODV protocol. Thus, the DLSMR+TD-PSO protocol with the RPGM model presented has the best performance which can improve connectivity and provide a stable connection compared to the other schemes regarding control overhead.

In Figure 14, we demonstrate the PDR as a function of multicast group size to evaluate the scalability of the DLSMR+TD-PSO protocol. It can be observed that the PDR has a nearly constant value and is unaffected when the multicast group size increases. The reason is that the DLSMR+TD-PSO protocol is capable of delivering packets to multiple destinations at the same time, ensuring consistent PDR even as the size of the multicast group increases. The protocol under the RPGM model shows superior performance compared to the RWP model. This is primarily because nodes in RPGM move in groups, which results in a more predictable and less chaotic network. Consequently, this leads to fewer packet losses and a better PDR, despite an increase in destinations. Furthermore, the DLSMR+TD-PSO protocol accurately predicts the secure next node during the routing process, even in a dynamic network. TD-PSO also helps the optimization of the allocation of network resources, thereby ensuring efficient data delivery. As a result, we can conclude that the DLSMR+TD-PSO protocol with the RPGM model has strong scalability in terms of PDR.

Then, we will turn our attention to look at the security performance. Figure 15 illustrates the comparison of the average packet loss ratio and number of packet losses in each session as a function of node speed as shown in Figure 15a and Figure 15b, respectively. According to Figure 15a, when the node’s speed increases, the ratio of packet loss increases. Furthermore, also in Figure 15b, when the node’s speed increases, the average number of packet losses increases. The reason is that, when the node’s speed increases, the node’s location frequently changes, thus causing packets to be transmitted directly to wormhole nodes. Furthermore, the proposed TD-PSO clustering protocol can reduce the number of links between nodes. Therefore, it can improve the packet loss ratio and the number of packet loss performances. On the other hand, the RPGM model instead of RWP can help the mitigation of the movement issue due to the collective and predictable movement of nodes resulting in fewer route changes and more stable connections, also reducing the likelihood of packet loss. Thus, the proposed DLSMR+TD-PSO protocol with the RPGM model is proven to be secure from a network security perspective.

Figure 16 shows the impact of the node density of the proposed DLSMR+TD-PSO protocol on the network metrics. We set the minimum number of nodes at 30, 50, and the maximum of 100. As can be seen in Figure 16a, the PDR decreases when the node speed increases, while the PDR slightly increases when the number of nodes increases. This behavior is influenced by the more predictable movement patterns in the RPGM model, which leads to less frequent route changes and fewer packet losses. Furthermore, we evaluate the routing delay as a function of node speed with different numbers of nodes, as shown in Figure 16b. It is observed that, when the number of nodes and speed of nodes increases, the routing delay increases slightly but not significantly. This means that when the number of nodes and node speed increases, the number of hops also increases, which can cause the routing process to take longer. In addition to that, the effect of node speed on the control overhead is illustrated in Figure 16c. It can be seen that when the node speed and the number of nodes increases, the control overhead increases. The reason is that the network density increases when nodes increase, resulting in more frequent packets to establish routes. Therefore, increased node speed or increased number of nodes requires more resources to establish a multicast route; consequently, the control overhead will be increased. Despite these challenges, it can be proved that the proposed DLSMR+TD-PSO protocol with RPGM mobility model has good scalability and effectively manages to improve the PDR, routing delay, and control overhead as the number of nodes increases.

## 6. Conclusions

In this paper, we proposed a DLSMR and TD-PSO protocol in FANETs with CF-mMIMO to establish a secure multicast route that improves node connectivity against wormhole attacks. The proposed DLSMR protocol utilized a DL approach to predict the secure and unsecured route based on various parameters such as node ID, distance, destination sequence, hop count, and energy which can avoid wormhole attacks. To enhance node connectivity, we proposed a TD-PSO clustering protocol that employed particle swarm optimization to find the global optimal points to maximize the cost function. This function considered the weight of the remaining energy, cosine similarity, cosine distance, and node degree, which led to electing the cluster head candidate. Furthermore, we also compared the protocol performance under two different mobility models (RPGM and RWP). The performance evaluation showed that the proposed DLSMR protocol with TD-PSO clustering protocol with the RPGM model can establish highly stable multicast trees that are robust to wormhole attacks. The proposed DLSMR protocol has better security performance than the MAODV protocol as a benchmark against wormhole attacks. This is indicated by the PDR and number of packet loss performance values which are better than MAODV. Simultaneously, the TD-PSO protocol can improve node connectivity and manage multicast members efficiently with good control overhead, number of cluster head changes, and routing delay. Overall, the proposed DLSMR protocol with TD-PSO clustering protocol under the RPGM model guarantees high stability, low routing delay, low packet loss ratio, low number of packet losses, low control overhead, and high PDR. To expand this work, we are trying to develop a secure routing protocol with a cross-layer design to protect the information against eavesdropper attacks using the DL technique. 

## Figures and Tables

**Figure 1 sensors-23-07960-f001:**
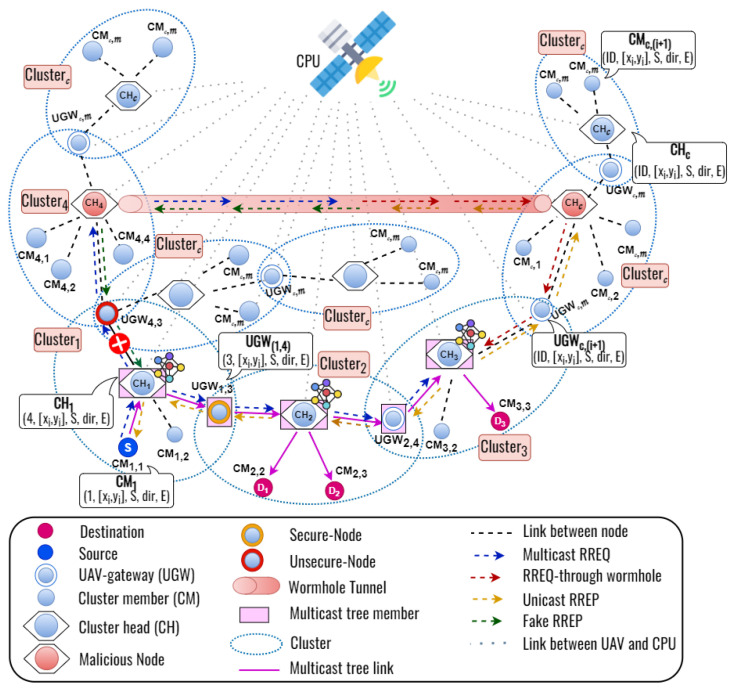
The basic concepts of the proposed DLSMR protocol.

**Figure 2 sensors-23-07960-f002:**
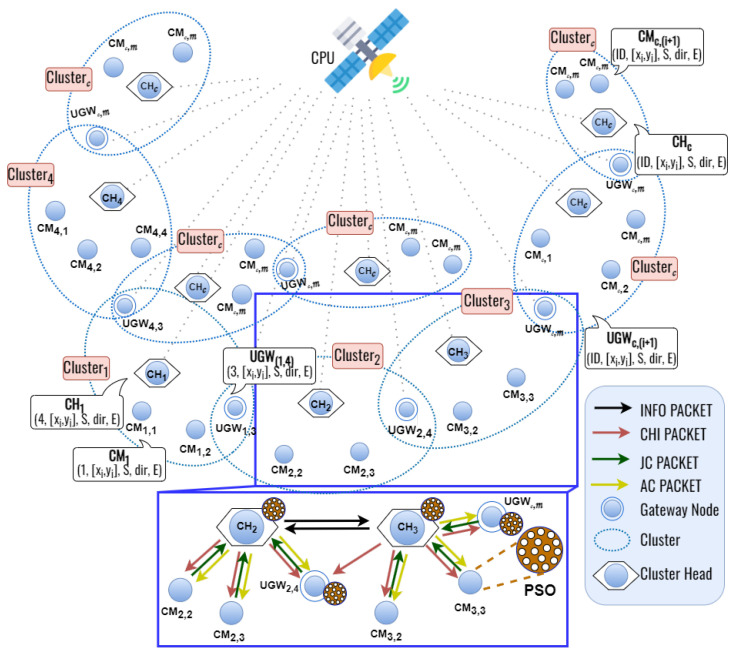
The basic concepts of the TD-PSO-based clustering protocol.

**Figure 3 sensors-23-07960-f003:**
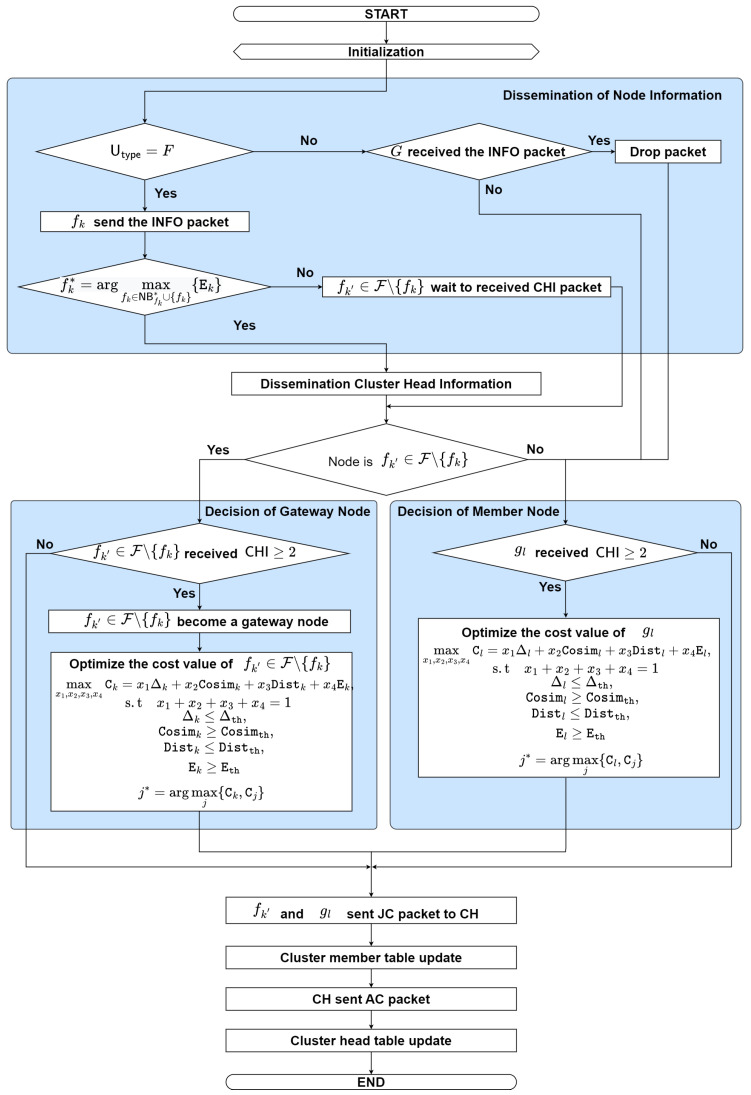
The flowchart of the proposed TD-PSO-based clustering protocol.

**Figure 4 sensors-23-07960-f004:**
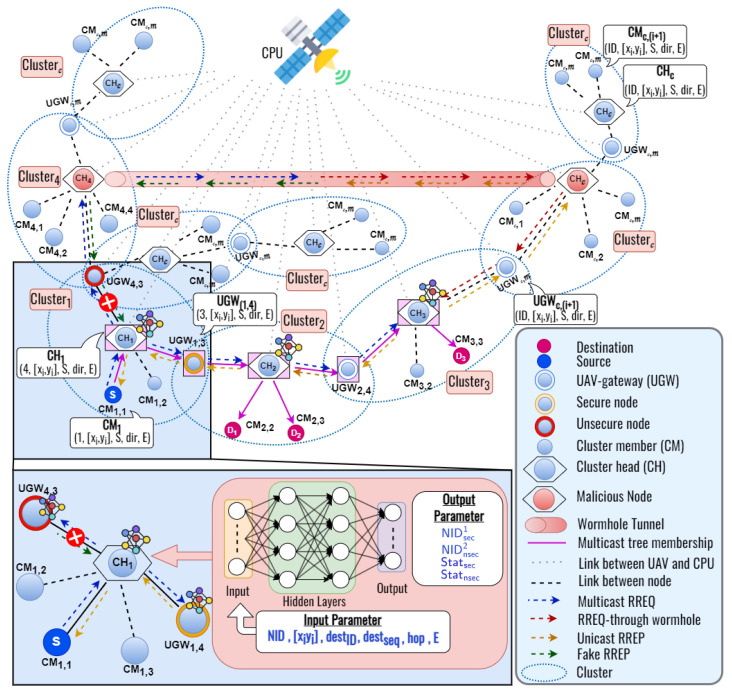
The basic concepts of the deep learning− (-) based secure multicast routing (DLSMR) protocol.

**Figure 5 sensors-23-07960-f005:**
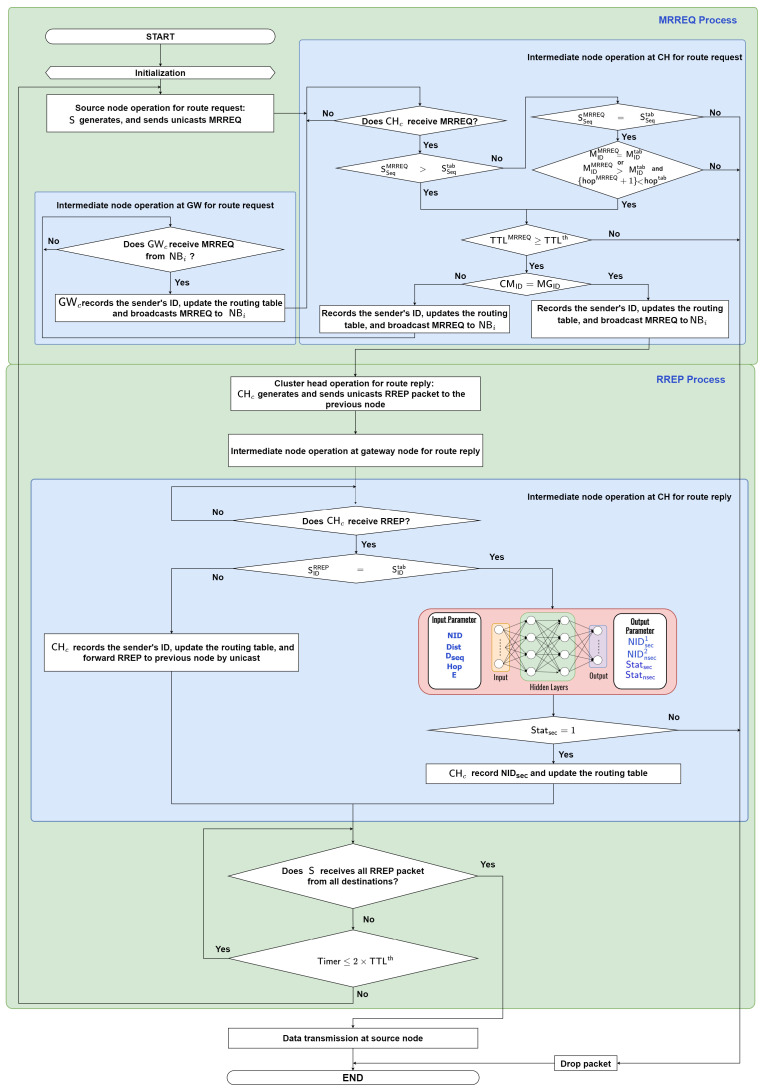
Flowchart of the proposed deep learning-based secure multicast routing (DLSMR) protocol.

**Figure 6 sensors-23-07960-f006:**
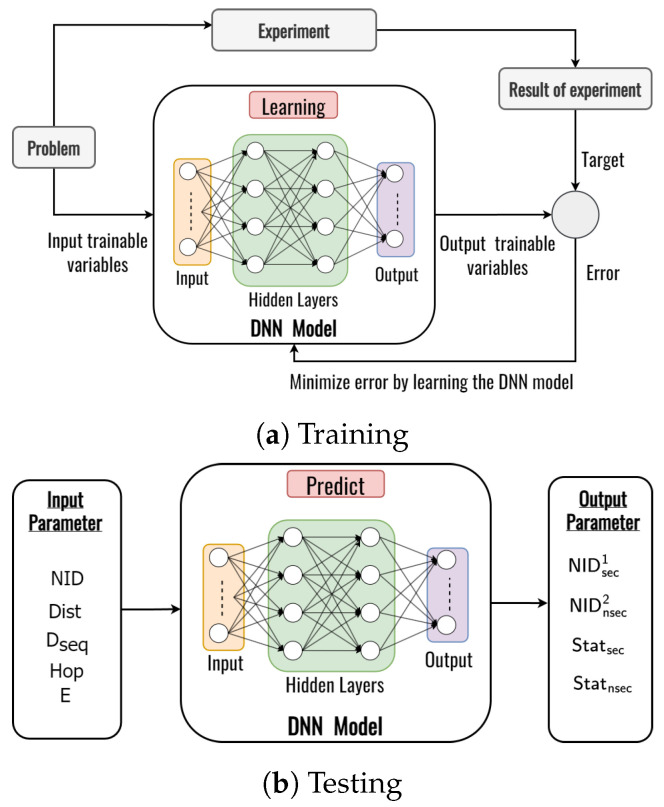
Comparison of packet loss ratio and number of packet losses as a function of node speed.

**Figure 7 sensors-23-07960-f007:**
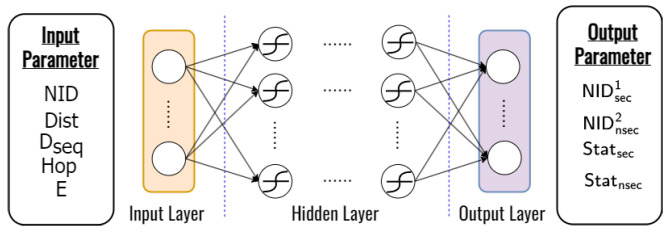
The structure of the deep learning framework for predicting the secure and unsecured next node.

**Figure 8 sensors-23-07960-f008:**
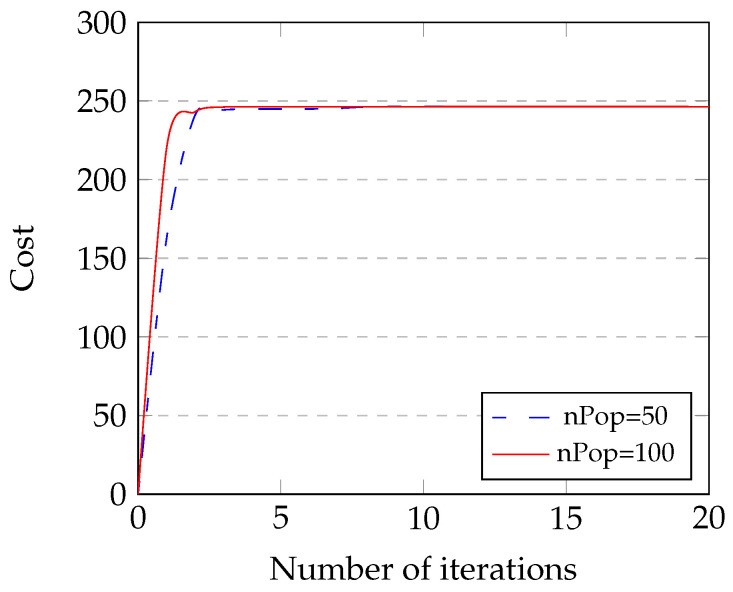
Convergence of Algorithm 1 for maximize cost problem as a function of the number of iterations.

**Figure 9 sensors-23-07960-f009:**
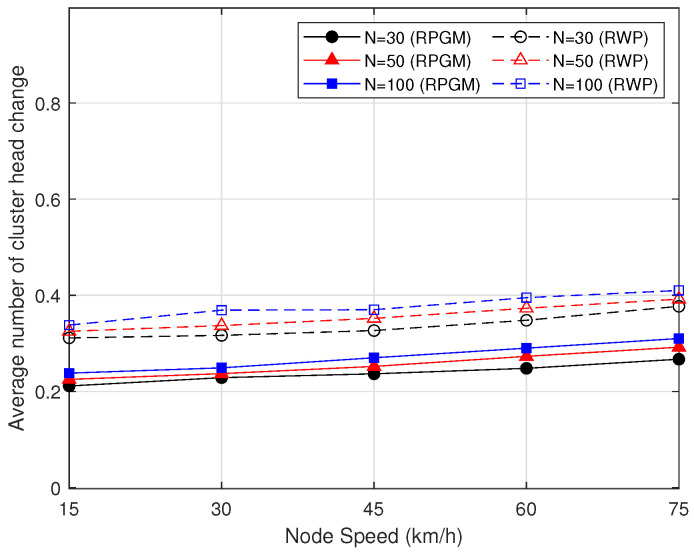
Average number of cluster heads changing as a function of node speed.

**Figure 10 sensors-23-07960-f010:**
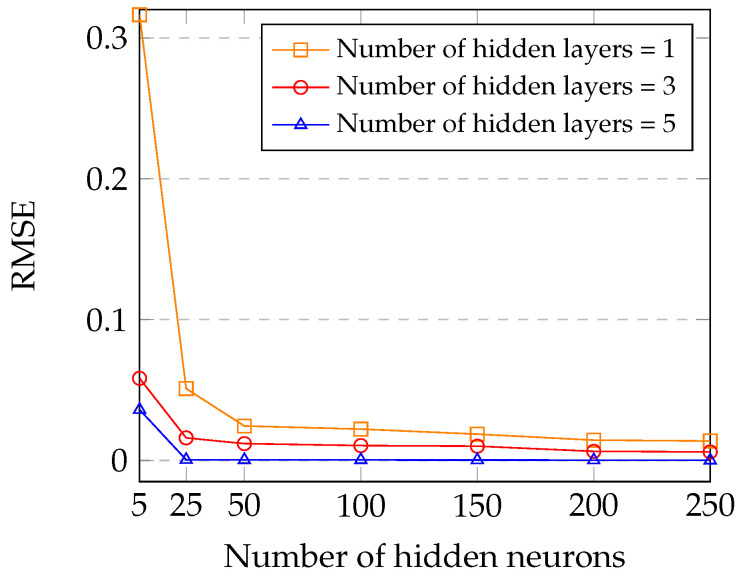
Impact of the number of hidden neurons on the DNN model with different numbers of hidden layers.

**Figure 11 sensors-23-07960-f011:**
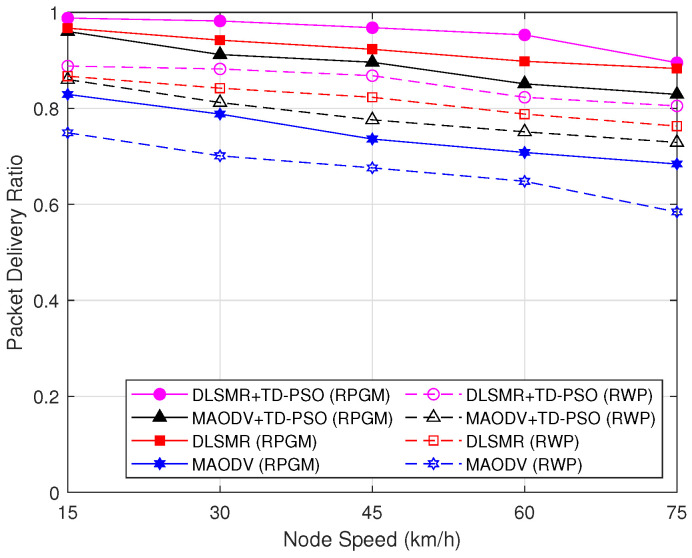
Packet delivery ratio as a function of node speed.

**Figure 12 sensors-23-07960-f012:**
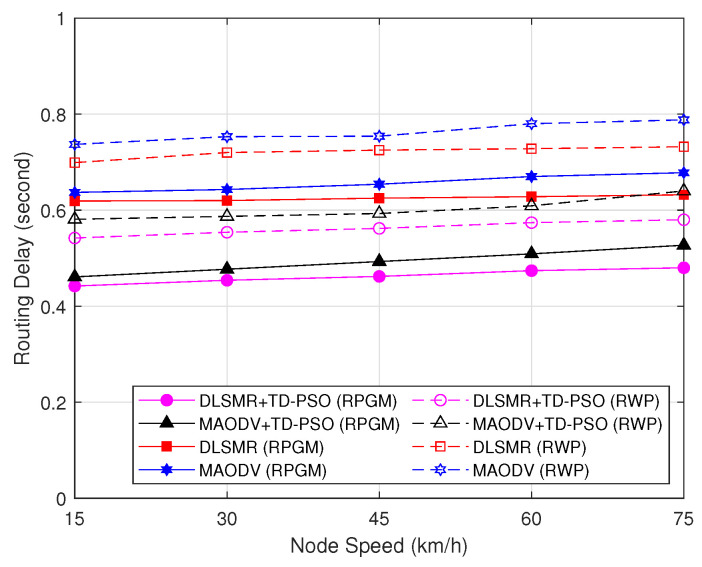
Routing delay as a function of node speed.

**Figure 13 sensors-23-07960-f013:**
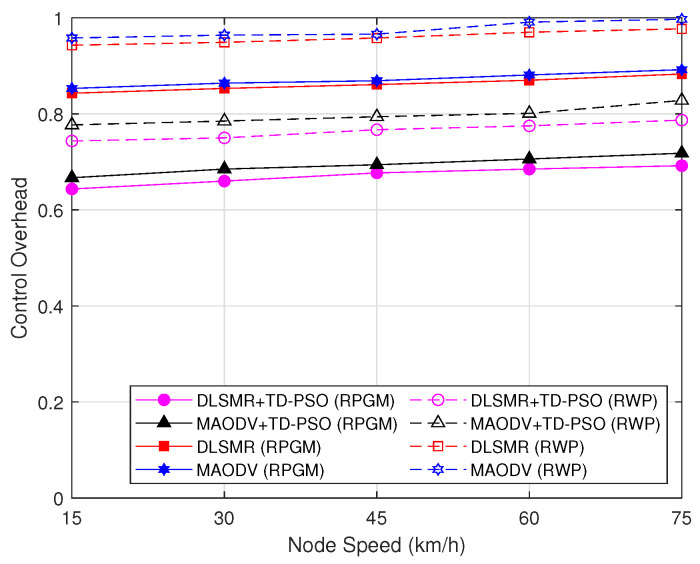
Control overhead as a function of node speed.

**Figure 14 sensors-23-07960-f014:**
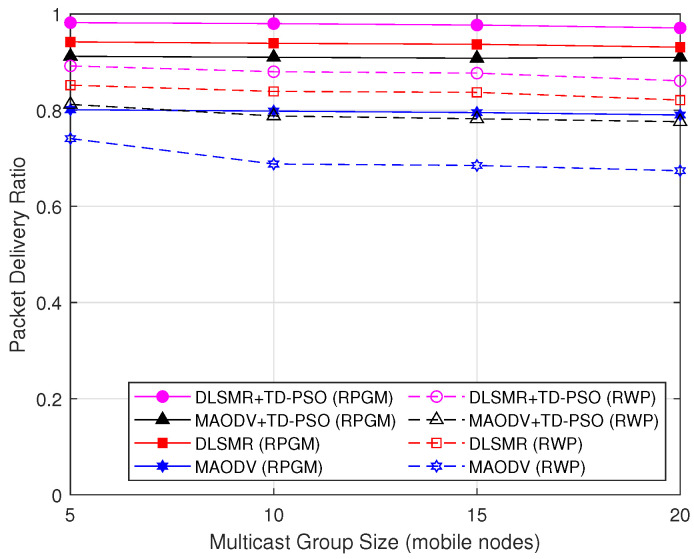
PDR as a function of multicast group size.

**Figure 15 sensors-23-07960-f015:**
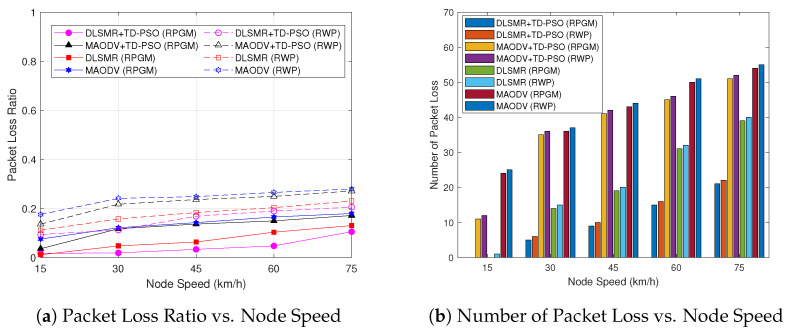
Comparison of packet loss ratio and number of packet losses as a function of node speed.

**Figure 16 sensors-23-07960-f016:**
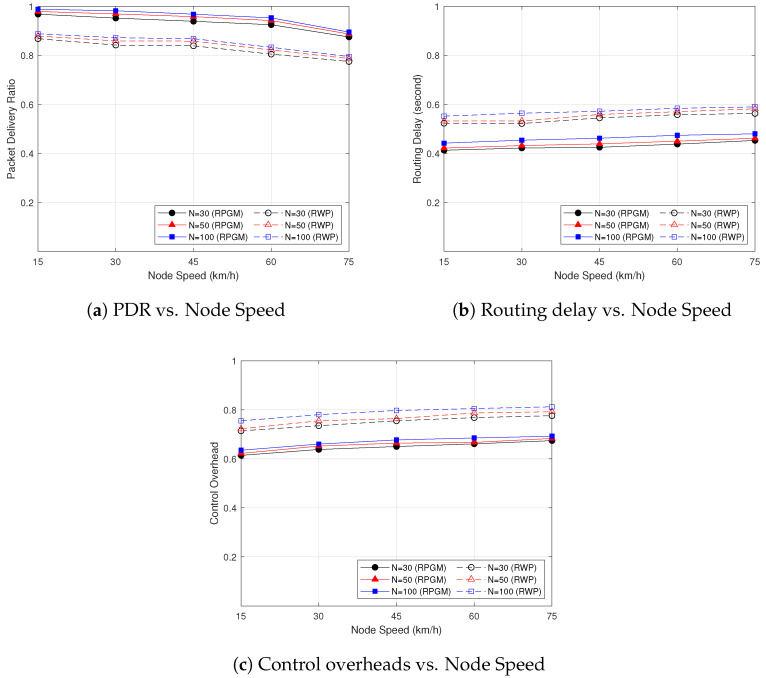
The impact of the number of nodes on the network performance metric.

**Table 1 sensors-23-07960-t001:** Cluster member table of the proposed TD-PSO-based clustering protocol.

${\mathrm{CM}}_{\mathrm{ID}}$	${\mathrm{CH}}_{\mathrm{ID}}$	$U_{\mathrm{type}}$

**Table 2 sensors-23-07960-t002:** Cluster head table of the proposed TD-PSO-based clustering protocol.

${\mathrm{CH}}_{\mathrm{ID}}$	${\mathrm{CM}}_{\mathrm{ID}}$	$U_{\mathrm{type}}$

**Table 3 sensors-23-07960-t003:** List of packets for the TD-PSO clustering protocol.

Packet Name	Full Name	Field Information
INFO	Information	$\mathrm{Type},S_{\mathrm{ID}},D_{\mathrm{ID}},E,U_{\mathrm{type}}$
CHI	Cluster Head Information	$\mathrm{Type},S_{\mathrm{ID}},D_{\mathrm{ID}},\mathrm{Pos},\mathrm{Dir},S,E,U_{\mathrm{type}}$
JC	Join Cluster	$\mathrm{Type},S_{\mathrm{ID}},D_{\mathrm{ID}},U_{\mathrm{type}}$
AC	Accept Cluster	$\mathrm{Type},S_{\mathrm{ID}},D_{\mathrm{ID}}$

**Table 4 sensors-23-07960-t004:** Multicast tree routing table of the proposed DLSMR protocol.

$S_{\mathrm{ID}}$	${\mathrm{MG}}_{\mathrm{ID}}$	$M_{\mathrm{ID}}$	$S_{\mathrm{Seq}}$	$D_{\mathrm{Seq}}$	${\mathrm{PN}}_{\mathrm{ID}}$	${\mathrm{NN}}_{\mathrm{ID}}$	Hop

**Table 5 sensors-23-07960-t005:** List of packets for the DLSMR protocol.

Packet Name	Full Name	Field Information
JREQ	Join Request	$\mathrm{Type},S_{\mathrm{ID}}^{\mathrm{JREQ}},D_{\mathrm{ID}},\mathrm{Status}$
MRREQ	Multicast Route Request	$\mathrm{Type},S_{\mathrm{ID}}^{\mathrm{MRREQ}},{\mathrm{MG}}_{\mathrm{ID}},M_{\mathrm{ID}},S_{\mathrm{Seq}},\mathrm{Hop},\mathrm{TTL}$
RREP	Route Reply	$\mathrm{Type},S_{\mathrm{ID}}^{\mathrm{RREP}},D_{\mathrm{ID}}^{\mathrm{RREP}},M_{\mathrm{ID}},D_{\mathrm{Seq}},E,\mathrm{Pos},\mathrm{Hop}$

**Table 6 sensors-23-07960-t006:** Model of the deep neural network to predict the secure next node.

	Size	Activation Function
Input	5	-
Layer 1	150	ELU
Layer 2	100	ELU
Layer 3	200	ELU
Layer 4	150	ELU
Layer 5	100	ELU
Output	4	LINEAR

**Table 7 sensors-23-07960-t007:** Parameters and simulation environments.

Parameters	Value
Simulator	NS-3 simulator
Simulation area	1000 × 1000 $m^{2}$
Packet size	1024 bits
Mobility model	RPGM and RWP
Transmission range	250 m
Simulation time	200 s
Session length	5 s
Number of nodes	[30, 50, 100]
Wormhole pairs	1 (Wormhole nodes 2)
Node’s speed range	[15:15:75] (km/h)
Receive signal strength indicator (RSSI) threshold	−80 dBm
MAC protocol	802.11a

**Table 8 sensors-23-07960-t008:** DNN training parameters.

Parameters	Value
Dataset	1,000,000
Epoch	50
Batch size	256
Optimizer	Adam
Initial learning rate	0.00001

## Data Availability

Not applicable.

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
