# Peer review of "DLSMR: Deep Learning-Based Secure Multicast Routing Protocol against Wormhole Attack in Flying Ad Hoc Networks with Cell-Free Massive Multiple-Input Multiple-Output†"

_sensors, 2023, doi:10.3390/s23187960_

Round 1

Reviewer 1 Report

The paper proposes a new secure multicast routing method against wormhole attack based on DL in FANET.  The effectiveness of the proposed technique is verified with simulation. However, the paper needs to address the following comments for publication.

 -          More description on recent studies on wormhole or other attacks against UAV networks should be provided in Section 1 Introduction as related works.

 -          A section describing wormhole attack model and PSO method should be added to help the readers’ understanding.

 -          Detailed design settings on DL and wormhole attack in Section 3 Performance Evaluation is missing.

 -          In Section 3 Performance Evaluation, other recently proposed secure UAV routing protocol against wormhole attack should be presented for better comparative analysis.

 -          In Conclusion, a deep insight on robustness of the proposed method against wormhole attack based on simulation results should be provided.

None.

Reviewer 2 Report

The following are comments to improve the paper:

1. The abstraction lacks a proper motivation. A brief overview of previous works and their shortcomings should be noted here. Additionally, the obtained improvement should be explicitly mentioned.

2. Use of "Can" in the abstract should be removed. Be-sure for for your proposed work.

3.  The introduction also needs a proper motivation. The previous works and their advantages and shortcomings should be added to this section. Additionally, add some references to the introduction. The beginning paragraphs of the introduction focus on general information, which should be summarized

4. List pointwise contributions in the abstract in third paragraph.

5. - In the conclusion, mention the summary of your experimental analysis in statistical form along with the future works.

Grammatical issues:
a. There are a lot of grammatical errors in the text. Some of them are highlighted. Please read the text carefully and fix the errors.
b. Don't omit "the" and "a" when it is necessary.

Round 2

Reviewer 1 Report

The authors have successfully addressed all the review comments. Therefore, I recommend publication of the submitted manuscript.

No comment.

Reviewer 2 Report

I have no more comments.